# Polystyrene as Graphene Film and 3D Graphene Sponge Precursor

**DOI:** 10.3390/nano9010101

**Published:** 2019-01-16

**Authors:** Alejandra Rendón-Patiño, Jinan Niu, Antonio Doménech-Carbó, Hermenegildo García, Ana Primo

**Affiliations:** 1Instituto de Tecnología Química, Consejo Superior de Investigaciones Científicas-Universitat Politécnica de Valencia, Av. De los Naranjos s/n, 46022 Valencia, Spain; Alrendon@upvnet.upv.es (A.R.-P.); jinan.niu@cumt.edu.cn (J.N.); 2Departament de Química Analítica. Universitat de València. Dr. Moliner, 50, 46100 Burjassot (València), Spain; antonio.domenech@uv.es

**Keywords:** graphene, polystyrene, 3D graphene sponges, electrochemistry

## Abstract

Polystyrene as a thin film on arbitrary substrates or pellets form defective graphene/graphitic films or powders that can be dispersed in water and organic solvents. The materials were characterized by visible absorption, Raman and X-ray photoelectron spectroscopy, electron and atomic force microscopy, and electrochemistry. Raman spectra of these materials showed the presence of the expected 2D, G, and D peaks at 2750, 1590, and 1350 cm^−1^, respectively. The relative intensity of the G versus the D peak was taken as a quantitative indicator of the density of defects in the G layer.

## 1. Introduction

Due to its wide availability and with the main objective of plastic wastes reutilization as carbon source, pyrolysis of polystyrene has attracted considerable attention [1,2]. Starting from the first report on 1950 [3,4,5,6], it was found that polystyrene decomposes in very high yields at temperatures above 370 °C, affording a complex mixture of volatile compounds [4,5,6,7,8,9]. Polystyrene pyrolysis can be carried out at convenient rates at temperatures of about 700 °C. Depending on temperature and operating conditions, the formation in significant amounts in variable proportions of toluene, ethylbenzene, cumene, styrene, and other benzene derivatives is observed [8,10]. 

Since the target of all these studies has been the use of polystyrene wastes as feedstock [1,11,12,13], the vast majority of the reports on polystyrene pyrolysis have focused on the analysis of the gas phase products, with the target being the obtainment of a suitable mixture to be used as fuel. Surprisingly, as far as we know, none of these studies have paid attention to the nature of the possible solid residue that could remain after pyrolysis. Evidence will be presented here showing that the residue of the polystyrene pyrolysis is a defective graphene. Transformation of polystyrene into graphenes can also be a way to valorize polystyrene wastes by converting them into high added-value products.

A few years ago, we reported that pyrolysis of natural polysaccharides considered to be food and agricultural wastes is a suitable process for the preparation of defective graphenes either as large-area films on arbitrary substrates [14] or as suspensions after sonication of the carbonaceous residue in liquid media [15]. About 40% of the mass of these polymeric carbohydrates are converted into carbonaceous residue. Pyrolysis has the advantage over chemical vapor disposition (CVD) that it can produce much larger amounts of graphenes in less time using less sophisticated equipment. On one hand, our contribution to this field was to show that these filmogenic polysaccharides form films on hydrophilic surfaces of appropriate substrates that upon pyrolysis are converted into continuous films of single-layer or few-layer defective graphenes [14]. On the other hand, it was shown that pyrolysis of carbohydrate particles as powders and subsequent sonication of the carbonaceous residue render dispersions of single- or few-layer defective graphene particles in almost quantitative exfoliation yield [15].

Although the procedure was also adapted to the preparation of doped and co-doped defective graphenes [16,17,18], it was not possible to decrease the oxygen content of the resulting graphenes below 10 wt %. The presence of residual oxygen is proposed to be due to the composition of the carbohydrates, which contain over 50 wt % in oxygen. After pyrolysis, a fraction of the initial oxygen content remained as oxygenated functional groups on the defective graphene sheet.

Continuing with alternative procedures for graphene preparation based on pyrolysis, either as films or as dispersible powders that could render graphene suspensions, it is of interest to explore the possibility to also use synthetic organic polymers that do not contain oxygen in their composition as precursors. In this way, the formation of graphene materials lacking oxygen in their composition could be possible. In addition, a well-known property of synthetic plastic polymers is their ability to form high-quality films on arbitrary supports which is a prerequisite to obtain graphene films.

Herein, it will be shown that polystyrene, either cast as films on arbitrary surface or as pellets, upon pyrolysis at 900 °C, forms single-layer or few-layer graphenes or graphitic residues that can be efficiently dispersed as single- or few-layer graphene or graphitic platelets (depending on the number of layers) in liquid media. The procedure can be adapted to the formation of tridimensional (3D) graphene sponges by coating using silica spheres of uniform particle size as hard templates.

## 2. Materials and Methods 

### 2.1. Preparation of Graphene Films

Polystyrene (Mw 280,000 amu by GPC, Aldrich, St. Louis, MO, USA) was dissolved in dichloromethane at a concentration of 3–30 mg/mL. The films were obtained by spin coating over 1 × 1 cm^2^ quartz or Copper substrate (APT-POLOS spin coater: 4000 rpm, 30 s). Polystyrene was pyrolyzed under argon atmosphere using the following oven program: heating at 5 °C/min up to 900 °C for 2 h. 

In an additional experiment, conversion of expanded polystyrene into graphene was performed by pyrolysis under argon atmosphere in a tubular oven heating at a rate of 5 °C/min up to 900 °C for 2 h. Film thickness before and after pyrolysis was measured in different parts of the sample by using a Veeco apparatus in taping mode. The thickness of the film was determined by making a scratch in the film with a cutter and monitoring the atomic force microscopy (AFM) profile perpendicular to the cut. Films on different substrates were measured. Note that the samples were not manipulated in a clean room and might contain dust. Also, the thermal stress caused some deformation on the substrate, therefore thickness measurements required the analysis of various parts of the films.

Raman spectra were acquired using a Labram-HR (Horiba Jobin Yvon, Gloucestershire, UK) Raman spectrometer (600 mm^−1^ grating, 100 mm entrance slit) coupled to a Peltier-cooled CCD detector (Andor DU420, Gloucestershire, UK) and an Olympus BXFM optical microscope. Raman scattering was produced by excitation at 514 nm by means of a HeNe laser with 0.1 mW excitation power on the sample. The laser beam was focused on the sample via a long-working distance X 50 microscope objective (numerical aperture = 0.5), which served also to collect the scattered photons. The Rayleigh scattering was removed by a holographic notch filter, and the Raman spectra were recorded between 200 and 2000 cm^−1^ with a resolution of 0.5 cm^−1^. Raman spectra were recorded from different parts of the film, allowing the samples to be mapped out.

### 2.2. Preparation of Graphene Sponges

One gram of monodispersed silica spheres (80 nm) was suspended in a solution of polystyrene in dichloromethane at a concentration of 50 mg/mL and stirred for 24 h under 40 °C reflux. This mixture was centrifuged at 4000 rpm for 5 min (Hettich Zentrifugen EBA 21, Tuttlingen, Germany) and washed twice with dichloromethane. After drying at 80 °C, the conversion of polystyrene into graphene was performed under argon atmosphere using an electrical furnace heating at a rate of 2 °C/min up to 900 °C for a holding time of 2 h. Then, the silica spheres were removed in a 2 M NaOH solution by stirring for 2 h at 80 °C. Finally, the product was collected by filtration.

### 2.3. Electrocatalytic Measurements

Cyclic voltammetric (CV) and electrochemical impedance spectroscopy (EIS) measurements were performed on graphene sheets deposited onto glassy carbon electrodes (geometrical surface area 0.071 cm^2^) in a conventional three-electrode electrochemical cell completed with an Ag/AgCl (3 M NaCl) reference electrode and a Pt wire auxiliary electrode using a CH 660I potentiostat. Air-saturated 0.10 M potassium phosphate buffer solution at Ph = 7 was used as a supporting electrolyte incorporating 0.5 mM K_3_Fe(CN)_6_ plus 0.5 mM K_4_Fe(CN)_6_ as a redox probe. EIS experiments were performed upon application of a sinusoidal potential modulation of ±5 mV amplitude in the 10^5^ Hz–10^−1^ Hz range. The bias potential was that of the equilibrium potential of the Fe(CN)_6_^3−^/Fe(CN)_6_^4−^ couple determined in CV measurements.

## 3. Results

### 3.1. Graphene Films

Initial experiments were carried out by casting polystyrene (average MW 280,000) onto copper foils or rigid transparent quartz substrates. Figure 1 illustrates the procedure of the formation of defective graphene films. It has been reported that pyrolysis of polystyrene converts almost quantitatively this polymer into volatile organic compounds [3]. In agreement to these precedents, thermogravimetric analysis under inert atmosphere of our polystyrene sample indicated a weight loss about 99%, confirming that most of the polymer had decomposed into volatile products.

In spite of an almost complete weight loss, and completing the previous studies analyzing volatile products, the characterization of the substrate surface after the pyrolysis of polystyrene films showed the presence on these surfaces of defective graphene films, in single layer, few layers, or multiple layers. In accordance with the expected mass loss and also with the tendency of graphene layers to form stacks, the pyrolytic process resulted in a considerable decrease in the thickness of the resulting graphene/graphitic films, which was reduced about three orders of magnitude compared with the initial polystyrene film thickness. In this way, after pyrolysis at 900 °C for 4 h under inert atmosphere, micrometric polystyrene films typically convert, as a rule of thumb, to nanometric, thick films corresponding to the stack of graphene layers. It has been reported that AFM is the best technique to determine the thickness of graphene films, from which the number of layers can be estimated [19]. As an example, Figure 2 shows AFM images of a polystyrene film precursor that is converted in the pyrolysis into a defective graphitic film of 7 nm thickness.

After pyrolysis, the films were also characterized by Raman and visible absorption spectroscopy. Figure 3c shows a representative example of the Raman spectra obtained for these defective graphene films. The Raman spectra of the graphene films showed the expected G band at 1590 cm^−1^ (full width at half height 150 cm^−1^) accompanied by a D peak at 1350 cm^−1^ (full width at half height 300 cm^−1^) that was associated with the presence of defects on graphene. The presence of a 2D peak at 2700 cm^−1^ on top of a broad background was also observed. According to the relative intensity of the 2D peak with respect to the G band, the film obtained corresponded to few-layer, defective graphene, or graphitic films [20]. This was in accordance with the AFM measurements, previously commented. The relative intensity of the G versus the D peak is usually taken as a quantitative indicator of the density of defects in the G layer [21,22]. In the present case, the I_G_/I_D_ ratio was about 1.19, which compares favorably with previous I_G_/I_D_ ratio values reported in the literature, for instance for reduced graphene oxides that are about 0.9 [23] or even for defective graphenes obtained by pyrolysis of natural polysaccharides (I_G_/I_D_ 1.15) [15].

For those films on quartz, optical transparency was in agreement with the number of layers of the resulting graphene films. As reported [14,24,25], the optical transmittance measured at long wavelengths (660 nm) can serve as a reliable and convenient procedure to determine the number of layers of samples. As an example, Figure 3 shows photographs of the polystyrene film on quartz before pyrolysis and the resulting defective few-layer graphene film after pyrolysis, as well as the relative transmittance in the UV–Vis region of three films having different thickness. In agreement with the calibration data reported in the literature [14,24,25], these films should correspond to single, double, and 10-layer graphene.

The nature of the carbon atoms in the defective graphene film was determined by deconvolution of the C1s peak in high-resolution X-ray photoelectron spectroscopy (XPS) (Figure 4). Survey XPS analysis showed the presence of C and O as the only detectable elements in the film. Deconvolution of the C1s for films of defective graphene on quartz substrates indicated that about 92% of the carbon atoms were graphitic carbons appearing at the expected 284.5 eV binding energy [26]. It was estimated that about 8% were sp^2^ C atoms bonded to oxygen with a binding energy value of 286 eV. These XPS data are compatible with the formation of defective graphene/graphitic films. It should be commented that, depending on the nature of the substrate, the percentage of C atoms bonded to oxygen can increase with respect to this value. In this way, using copper foils as substrates, defective graphene films exhibited very similar Raman spectra as those presented in Figure 3; however, in XPS the C1s peak was broader, more asymmetric, shifted towards higher binding energies and presented a significantly larger contribution of C atoms bonded to O at 286.5 eV and even the presence of carboxylic C atoms in 6.4% at 289.0 eV. Since the oxygen content of polystyrene precursor was negligible and the pyrolysis was carried out under inert atmosphere, the XPS data suggests that, during the pyrolytic process, the surface of the substrate was providing some oxygen to the nascent defective graphene that became incorporated into the defective graphene. It should be remarked that no special precautions for the presence of surface oxides were taken in the case of copper foils. It also appears that quartz seems more reluctant than copper to act as an oxygen donor during pyrolysis. This seems to be in accordance with the chemical nature of copper that is more reducible in comparison with silicon.

Scanning electron microscopy (SEM) images of the defective graphene films after pyrolysis showed smooth continuous films without formation of cracks and crevices at micrometric length scale (Figure 5a). Upon detaching some material from these films, transmission electron microscopy (TEM) images of the detached film could be taken. These images from the detached films showed the expected layered morphology of the material and the absence of amorphous carbon regions. High-resolution TEM images clearly showed the structural ordering at the nanometric scale characteristic of graphene with domains in which the orientation of the lattice changes (Figure 5).

Films on quartz of defective few-layer graphene obtained from polystyrene exhibited a notable electrical conductivity in the range below kΩ/square, typically about 170 and 270 Ω/square. This electrical conductivity was sufficiently high to allow the recording of high-resolution scan tunneling microscopy (STM) of the samples. Appendix A (Appendix A) presents an illustrative example of the images taken of these defective graphene films, showing the presence of atoms in the expected arrangement. Unfortunately, the lack of sufficient flatness of the quartz substrate, particularly after being submitted to the thermal stress of the pyrolysis process, precluded recording the image of a large area of these films. It should be noted in this context that previous attempts to record STM images of defective graphene films from chitosan and alginate pyrolysis met with failure, due to their insufficient electrical conductivity. In this regard, the behavior of polystyrene-derived graphene films is remarkable compared to precedents in the literature that report STM for high-quality graphene films prepared by chemical vapor deposition on facet oriented clean metal surface [27].

Besides films, polystyrene was also pyrolyzed as pellets. It was observed that during the initial stages of temperature increase, polystyrene pellets melted and coated the ceramic crucible used in the pyrolysis process forming spontaneously a high-quality film. If the amount of polystyrene in the crucible was large, films of several millimeters thickness could be obtained due to the melting of the plastic. At the end of pyrolysis, bright metallic carbon residues coating the ceramic crucible were observed. For thick films, the carbonaceous residue could be recovered from the crucible by scratching, and the solid residue was suspended by sonication in various solvents. Figure 5 also provides some TEM images of the graphene material present in the suspensions, whereby the typical hexagonal atomic arrangement characteristic of graphene can also be observed.

The films of defective graphenes on glassy carbon electrodes were characterized electrochemically by EIS and CV using Fe(CN)_6_^3−^/Fe(CN)_6_^4−^ couple as a redox probe. EIS of defective graphene films is shown in Figure 6. The experimental data were satisfactorily modeled using the Randles-type equivalent circuit (see inset in Figure 6) composed by solution resistance (R_s_) in series with a parallel combination of a constant phase element (Q_dl_), representative of the non-ideal capacitance at the electrolyte–sheet interface with a branch containing a resistance (R_ct_) representative of the charge-transfer resistance at the above interface, in series with a Warburg element (W), representative of the existing diffusive effects. This equivalent circuit is the same as that proposed for describing the EIS of graphene oxide sheets on glassy carbon electrode [28,29,30]. The *R*_s_, *R*_ct_, *Q*_dl_, and *W* values determined from the EIS of these defective graphene films were 200 Ω, 7900 Ω, 6.8 × 10^−5^ Ω s^−n^ (n 0.92), and 2.3 × 10^−3^ Ω × s^−1/2^, respectively. 

Cyclic voltammograms of the Fe(CN)_6_^3−^/Fe(CN)_6_^4−^ redox pair at the defective graphene film on glassy carbon displayed the characteristic peaks for the essentially reversible interconversion of the two species (Figure 6), as judged by the cathodic-to-anodic peak potential separation tending to 60 mV at potential scan rates below 5 mV s^−^^1^. The peak-to-peak separation, however, increased with an increasing scan rate, as expected for systems combining uncompensated ohmic losses accompanying small deviations from reversibility. The process at both graphene-modified and unmodified electrodes was diffusion-controlled, as evidenced by the proportionality between the peak currents and the square root of sweep rate between 1 and 500 mV s^−^^1^. The background current (dotted lines in Figure 6) approached the rectangular-type shape characteristic of graphene-modified electrodes, being representative of the appearance of significant capacitive effects. As can be seen in Figure 6, the peak current at graphene-modified electrodes increased significantly with respect to the value recorded at unmodified glassy carbon electrodes, thus indicating the enhancement of the effective electrochemical area produced by the graphene film. No evidence for additional voltammetric signals or distortions between the anodic and cathodic profiles of the voltammograms were detected, thus suggesting that no carboxylation of the graphene occurred.

### 3.2. 3D Graphene Sponges

Besides forming flat films, the possibility to exploit the filmogenic ability of polystyrene to adopt the form of the substrates for the formation of 3D graphene sponges was considered. The structuring of graphene into 3D objects is important as a way to increase the surface area of the material, for instance in the preparation of electrodes and supercapacitors, among other possible applications [31,32,33,34]. A general strategy to obtain 3D graphene sponges with regular pore size is the use of hard templates [35]. Silica spheres of uniform diameter are among the preferred hard templates to form 3D sponges [36]. In some procedures, graphene was formed by chemical vapor deposition on silica spheres as templates [37], or graphene oxide was adhered to silica spheres and used as precursor of reduced graphene oxide sponges [35]. 

In the present study, silica spheres of uniform dimensions of about 80 nm were prepared by the Stöber hydrolysis procedure and used as templates [38]. These silica spheres were impregnated with polystyrene by stirring a suspension of both components in CH_2_Cl_2_. After recovery of the impregnated spheres and washing to remove polystyrene excess, pyrolysis of the impregnated silica spheres was carried out at 900 °C under N_2_ atmosphere. The silica spheres were removed by etching of the resulting carbonaceous composite material with NaOH. Figure 7 summarizes the procedure for the preparation of the 3D defective graphene sponges.

Raman and X-ray photoelectron spectroscopy of the 3D defective graphene sponges were mostly coincident with those previously commented. Interestingly, SEM images of the 3D porous graphene sponges after removal of the silica spheres showed the presence of a regular porosity in the material. Figure 8 shows some selected images at different magnifications to illustrate the morphological features of the 3D graphene sponges prepared from polystyrene.

As observed in these images, the 3D graphene sponge morphology is constituted by hexagonal cavities of very uniform dimensions of about 80 nm. This dimension is commensurate with the diameters of the silica spheres used as templates. These cages have four windows of about 40 nm diameter that are tetrahedrally arranged. The thickness of the wall is about 5 nm which is consistent with them being constituted by few-layers defective graphene.

## 4. Conclusions

Although pyrolysis of graphene has been exhaustively studied as a way to convert this synthetic polymer into fuels, no attention has been paid to the possibility to form graphene. In the present manuscript, it is shown that pyrolysis of polystyrene films or pellets form defective graphenes either as films or as residues that can be dispersed in liquid media. Depending on the nature of the substrate, the resulting graphene can incorporate oxygen in a variable percentage. The notable electrical conductivity of these defective graphene films allows the monitoring of the surface by scanning tunneling microscopy and the us of these films as electrodes. The filmogenic properties of polystyrene also make possible the preparation of 3D graphene sponges with remarkable uniform pore size by using silica spheres as hard templates. The present finding paves the way for the preparation of doped defective graphene and heterojunctions by taking advantage of the ability of polystyrene to form these graphenes.

## Figures and Tables

**Figure 1 nanomaterials-09-00101-f001:**
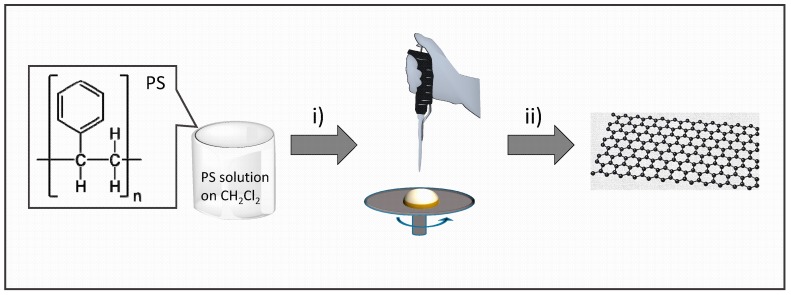
Illustration of the preparation procedure of defective graphene films by pyrolysis of polystyrene. (i) spin coating of clean substrate with a CH_2_Cl_2_ solution of polystyrene; (ii) pyrolysis under N_2_ at 900 °C for 2 h.

**Figure 2 nanomaterials-09-00101-f002:**
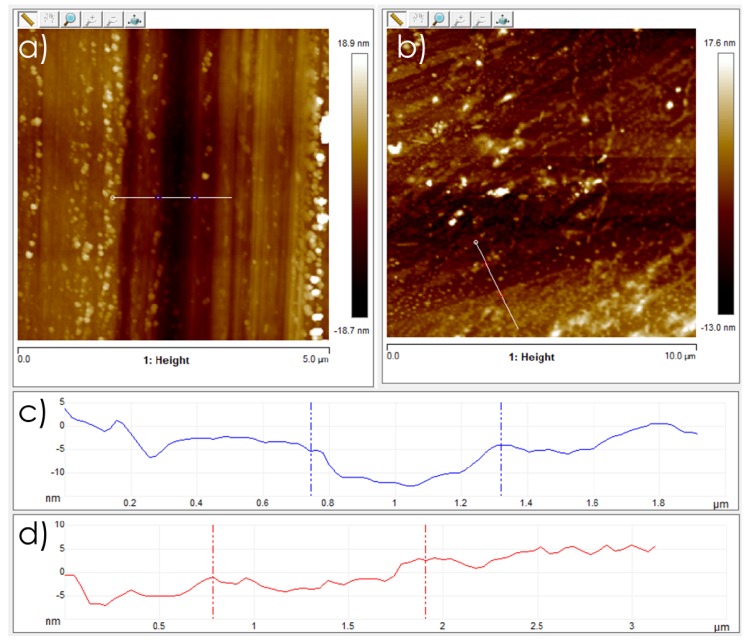
AFM images of (**a**) polystyrene film precursor with a measured thickness of 15 nm and (**b**) defective graphene film from pyrolysis of the previous polystyrene film with a measured thickness of 7 nm. Note that panels (**c**,**d**) may not correspond to the same location.

**Figure 3 nanomaterials-09-00101-f003:**
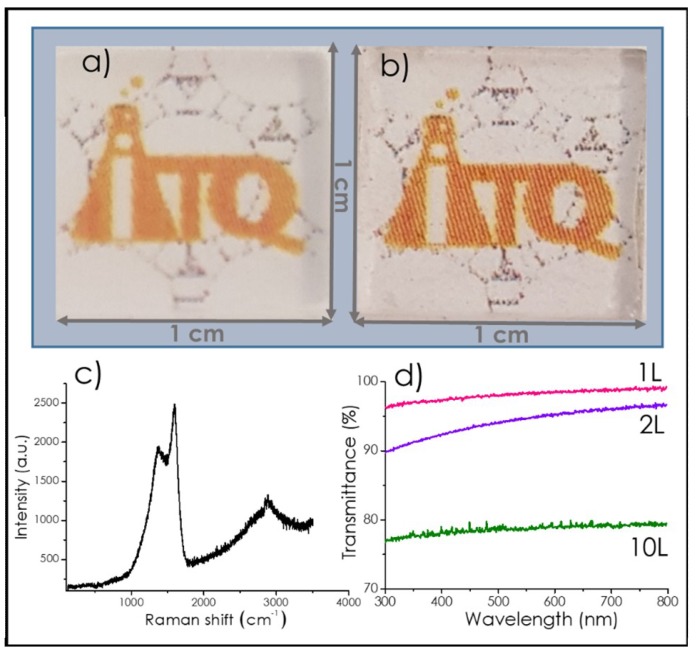
Photograph of quartz support with (**a**) PS film before pyrolysis and (**b**) graphene film after pyrolysis of PS; (**c**) Raman spectrum recorded upon 514 nm excitation for defective graphene films on quartz obtained by pyrolysis of polystyrene films; (**d**) Optical transmittance vs wavelength for 1 layer graphene film ([PS] = 3 mg/mL), 2 layers graphene film ([PS] = 10 mg/mL), 10 layers graphene film ([PS] = 30 mg/mL), spin coating rate 4000 rpm.

**Figure 4 nanomaterials-09-00101-f004:**
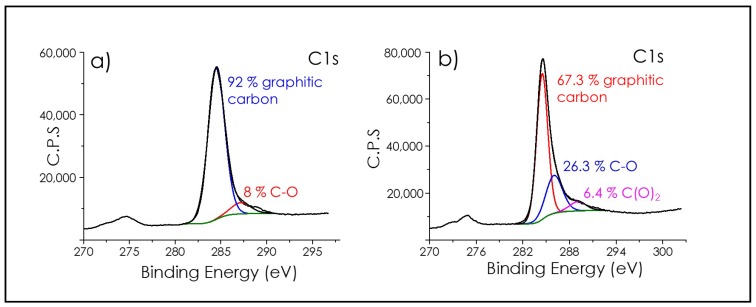
Deconvolution of the C1s peak in high resolution XPS of (**a**) defective graphene on quartz support and (**b**) defective graphene in Copper foil.

**Figure 5 nanomaterials-09-00101-f005:**
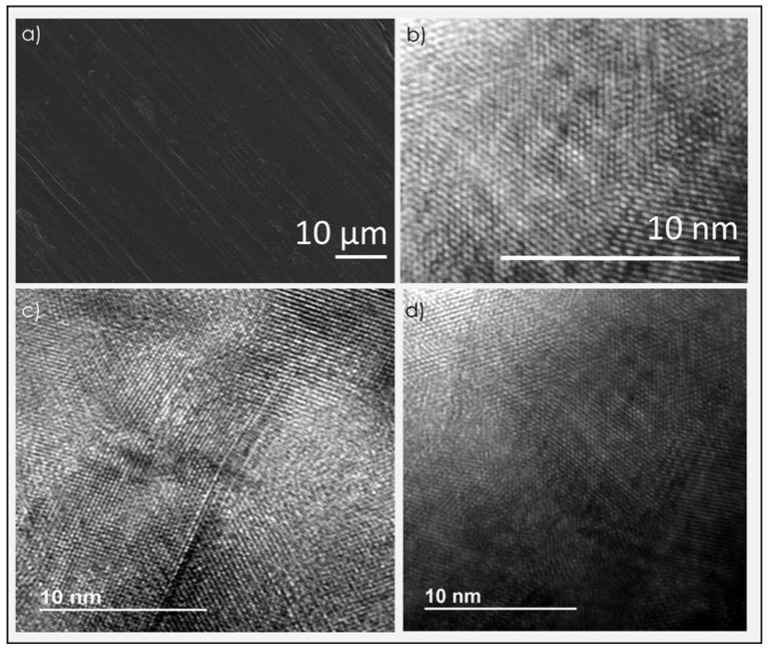
(**a**) SEM images and (**b**) high resolution TEM images of the defective graphene films after pyrolysis and (**c**,**d**) High resolution TEM images of defective graphene from polystyrene pellets.

**Figure 6 nanomaterials-09-00101-f006:**
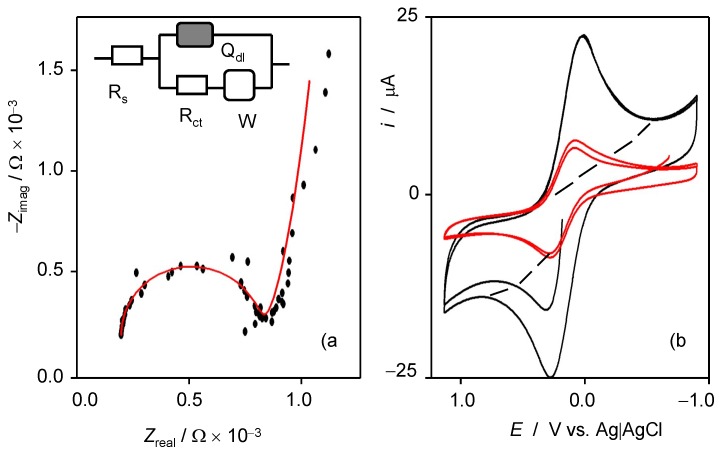
Electrochemical characterization of graphene films deposited on glassy carbon electrode in contact with 1.0 mM K_3_Fe(CN)_6_/K_4_Fe(CN)_6_ solution in 0.10 M potassium phosphate aqueous solution at pH 7.0. (**a**) Nyquist plot of EIS at a bias potential of 0.20 V vs. Ag/AgCl; (**b**) CVs at the unmodified (red) and graphene-modified (black) electrode, potential scan rate 20 mV/s. The dotted line represents the background current in the absence of K_3_Fe(CN)_6_/K_4_Fe(CN)_6_. The inset in Figure 6a corresponds to the equivalent circuit providing the theoretical impedance spectrum depicted as the continuous line in the same.

**Figure 7 nanomaterials-09-00101-f007:**
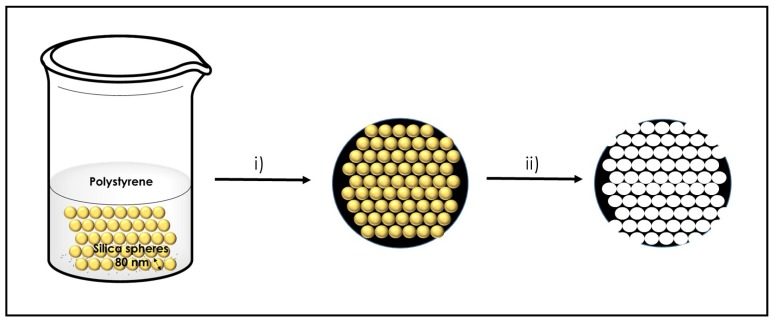
Illustration of the preparation procedure of 3D defective graphene sponges.

**Figure 8 nanomaterials-09-00101-f008:**
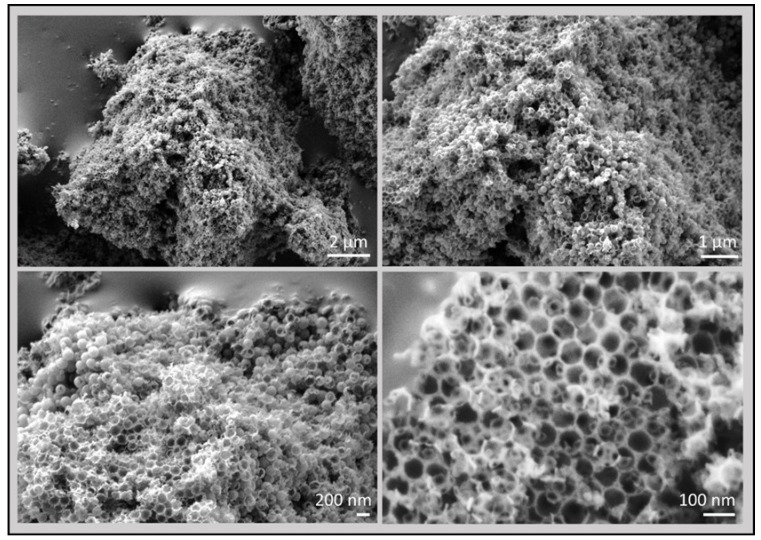
SEM images at different magnifications of 3D graphene sponges prepared from polystyrene.

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
