# Peer review of "Polystyrene as Graphene Film and 3D Graphene Sponge Precursor"

_nanomaterials, 2019, doi:10.3390/nano9010101_

Round 1
Reviewer 1 Report
The authors have revised the ms taking into consideration the referees' suggestions. I'm happy with the amendments been made and I thibnk the ms is acceptable for publication.
Reviewer 2 Report
This referee recommends the paper is accepted.
Reviewer 3 Report
Authors have tried to reply to my and other reviewers' comments. Since they have at least added the term graphitic I recommend publication because I understand that I can't convince them to delete the term graphene. The Raman they present has nothing to do with the Raman of graphene they have cited. By the way, I referred to Prof. Ferrari regarding Raman not to cite him but to read carefully his publications about the Raman of carbon materials. In addition, I am far from convinced also regarding the optical transparency and the uniformity of the films...
Reviewer 4 Report
The paper is now accepted for publication.
This manuscript is a resubmission of an earlier submission. The following is a list of the peer review reports and author responses from that submission.
Round 1
Reviewer 1 Report
The ms describes a synthetic procedure for the fabrication of graphene sponge precursor.
The ms is interesting and well written and I recommend acceptance after revision.
1) I wonder if the material can be defined as graphene or as graphite platelets. There is now a convention in order to univocally define carbon nanomaterials;
2) the electrochemical characterisation is minimal. The CV should report the background current as well. From the shape of the CV the graphene has a high capacitive current and this should be explained in more detail in the text. Also what is the peak separation? is the process diffusion controlled? this section should be improved.
3) from the electrochemical point of view, is the graphene carboxylated or else?
Reviewer 2 Report
This reviewer has concerns over the experimental details provided and the overall analysis of the data with particular focus on the AFM and Raman data presented. This reviewer would like to see the following questions /suggestions answered before acceptance of the article can be recommended
1. Figure 2 shows AFM of a polystyrene film and a graphene film. Why are the images shown amplitude error images ? The vertical scale for amplitude error images are given in V or mV. The authors should display height images not amplitude error images.
2. In figure 2a there are obvious tip artefacts on the left hand side of the film. Please comment on this so that readers are aware of this in the image. What was the approximate thickness of the polystyrene films measured using AFM ?
3. In figure 2b the authors claim they have formed a graphene film. The AFM image shows what looks like particles rather than any layers or flakes of graphene. Please comment. The authors also state that the graphene film is 5 nm thick in the caption of figure 2. 5 nm is not few layer as the authors claim but is likely over 10 layers or more. Please comment. The authors should also refer to and cite the following article that covers the AFM of graphene in great detail
Accurate thickness measurement of graphene. CJ Shearer, AD Slattery, AJ Stapleton, JG Shapter, CT Gibson. Nanotechnology 27 (12), 125704 (2016).
4. The claim of 5 nm graphene film thickness is based on how many measurements? This referee hopes more than one. Please do more details analysis of the data and please provide more experimental details overall. How many AFM images were acquired per sample? What was the AFM used? What types of AFM probes were used? What are the probes nominal properties? What was the imaging mode used to acquire the AFM images? How were the AFM images analysed? How as the AFM scanner calibrated? Please answer all these points and provide more information.
5. The cross sections under the AFM images look like screen dumps form the AFM software. Please export these as text files and re-plot these in a drawing package so they look more presentable.
6. The Raman spectra shown in figure 3c does not seem very impressive. How do the authors know this is graphene and not graphite ? The ratio of the peak height for the 2D band compared to the G band can give information on this as well as the shape of the 2D band. Please read the literature to support your claims concerning this data.
7. Again not enough information is given regarding the experimental details. What type of Raman instrument was used ? What was the power levels ? What was the objective used ? What was the spectral grating ? What was they integrations time ? Why did you not do any background subtraction to the data in figure 3c ? How many Raman spectra were acquired per sample ? Again I hope more than one.
8. In figure 3d the authors claim they are working with 1,2 and 10 layer graphene films. Where is the evidence for this? Raman spectra should be shown to verify these claims.
9. The spacing between graphene layers is well known to be 0.34 nm. Why is this not shown in the TEM images in figure 5 ? This should be made very clear to the readers.
Reviewer 3 Report
This work claims the preparation of defective graphene films and sponges by pyrolyzing polystyrene. I have many doubts that this is indeed defective graphene and not amorphous carbon with a few islands of higher and graphitic crystallinity.
Basically, Raman spectroscopy is probably the best tool to recognize and distinguish carbon structures. With this Raman, I can’t say that this is a defective graphene but rather an amorphous carbon with higher crystallinity islands. Which in fact, is more probable to get by this method. Besides, the broadening of all three peaks implies amorphous structure and not graphene, whereas the D to G peak intensity ratio shows nothing (or is totally misleading if you don’t know what exactly carbon material you have). Please, read the publications of Prof. Andrea Ferrari who is an expert in Raman spectroscopy of graphene, amorphous carbon and related materials. In addition, you write that the G peak is around 1590 cm-1. What is the exact position? You don’t mention anything about the measurement setup except from the wavelength used. Have you checked the instrument for possible shifting before measurements?
In addition, the rest techniques don’t show that you have FLG films as you claim.
AFM images and height analysis show nothing.
You write: “Scanning electron microscopy images of the defective graphene films after pyrolysis show a smooth continuous films at the submillimetric length scale (Figure 5a).” I can’t see that.
Apparently, you can’t claim to have graphene from any XPS.
Figure 5c: Doesn’t this seem like amorphous carbon with higher crystallinity islands?
Furthermore, is pyrolysis under argon as written in materials and methods section or under nitrogen as written in the caption of Figure 1?
Are AFM images of Figure 2 at the same spot? And basically, what I concern about regarding this, is the following. You write about a general rule that a microns-sized precursor film should give a nanometric graphene film, but you have no citations there. So, is this something that you claim here? It is rational that here this is pretty much the case for the heights but if this is a rule (I mean the three orders of magnitude you mention), would this mean that putting a ~300 nm thick polystyrene film one would expect a single layer graphene?
Visible spectra and analysis are misleading. Number of graphene layers can be correlated with T% but not for such kind of films but for dispersions or truly uniform films (CVD grown, etc.). You already presented an AFM image showing that you don’t have continuous films so much of your T% may originate from clean spots. You cannot claim the number of layers while you don’t mention anything about the coverage.
Overall, I can’t understand why it has to be graphene? You did some experiments, you got some results, why you have to claim that you get graphene?
Reviewer 4 Report
In the present manuscript, the pyrolysis of polystyrene films or pellets forming defective graphenes is presented either as films or as residues that can be dispersed in liquid media. The authors of this manuscript showed in a well-organized manner that depending on the nature of the substrate, the resulting graphene can incorporate oxygen in a variable percentage presenting electrical conductivity. The materials were well characterized by visible absorption, Raman and X-ray photoelectron spectroscopy, electron and atomic forcemicroscopy as well as electrochemistry.
I believe the findings of this work are quite important and can shed light to the preparation of doped defective graphene and heterojunctions by taken advantage of the ability of polystyrene to form the starting graphene material and therefore I recommend it to be accepted for publication.
As a minor addition/correction to the manuscript I suggest the authors to extent the introduction part describing in more detail about polystyrene wastes as feedstock and alternative procedures for graphene preparation based on pyrolysis.